# Vitamin B12 Status before and after Outpatient Treatment of Severe Acute Malnutrition in Children Aged 6–59 Months: A Sub-Study of a Randomized Controlled Trial in Burkina Faso

**DOI:** 10.3390/nu15163496

**Published:** 2023-08-08

**Authors:** Victor Nikièma, Suvi T. Kangas, Cécile Salpeteur, André Briend, Leisel Talley, Henrik Friis, Christian Ritz, Ebba Nexo, Adrian McCann

**Affiliations:** 1Expertise and Advocacy Department, Action Against Hunger (ACF), 93558 Montreuil, France; csalpeteur@actioncontrelafaim.org; 2Airbel Impact Lab, International Rescue Committee, New York, NY 10168, USA; suvi.kangas@rescue.org; 3Department of Nutrition, Exercise and Sports, University of Copenhagen, 1958 Copenhagen, Denmark; andre.briend@gmail.com (A.B.); hfr@nexs.ku.dk (H.F.); ritz@sdu.dk (C.R.); 4Tampere Center for Child, Adolescent and Maternal Health Research, Faculty of Medicine and Health Technology, Tampere University and Tampere University Hospital, 33100 Tampere, Finland; 5Centers for Disease Control and Prevention, Atlanta, GA 30329, USA; lre0@cdc.gov; 6Department of Clinical Biochemistry, Aarhus University Hospital, 8200 Aarhus, Denmark; enexo@clin.au.dk; 7Bevital AS, 5021 Bergen, Norway; adrian.mccann@bevital.no

**Keywords:** children, severe acute malnutrition, ready-to-use therapeutic foods, serum B12, combined indicator of B12 status, Burkina Faso

## Abstract

Severe acute malnutrition (SAM) is treated with ready-to-use therapeutic foods (RUTF) containing a vitamin–mineral premix. Yet little is known about micronutrient status in children with SAM before and after treatment. We aimed to investigate vitamin B12 status in children with uncomplicated SAM, aged 6–59 months in Burkina Faso, before and after treatment with a standard or a reduced dose of RUTF. Blood samples were collected at admission and discharge. Serum B12 was determined with microbiological assay and serum methylmalonic acid (MMA) and total homocysteine (tHcy) were analyzed with gas chromatography-tandem mass spectrometry. B12 status was classified using the combined indicator (3cB12). Among 374 children, the median [interquartile range] age was 11.0 [7.7–16.9] months, and 85.8% were breastfed. Marked or severe B12 deficiency, as judged by 3cB12, decreased from 32% to 9% between admission and discharge (*p* < 0.05). No differences in B12 status following treatment with either standard (*n* = 194) or reduced (*n* = 180) doses of RUTF were observed. Breastfed children showed a lower B12 status (3cB12) than non-breastfed ones (−1.10 vs −0.18, *p* < 0.001 at admission; −0.44 vs 0.19; *p* < 0.001 at discharge). In conclusion, treatment of SAM with RUTF improved children’s B12 status but did not fully correct B12 deficiency.

## 1. Introduction

Vitamin B12 (cobalamin) is of major importance for child development and growth [1]. Yet, to our knowledge, no study has investigated B12 status in children with severe acute malnutrition (SAM) and whether B12 biomarker concentrations improve following treatment with ready-to-use therapeutic foods (RUTF). RUTF are used for treating millions of children with SAM every year; they are fortified with a vitamin-mineral premix which has been formulated based on expert opinion with little evidence to support its composition [2].

Vitamin B12 is essential for DNA, RNA, and protein synthesis; DNA methylation; myelination of neurons and, hence, brain development [3], particularly during the first 1000 days of life [4,5]. During infancy and childhood, B12 deficiency can result in anemia, poor growth, and increased risk of infection, and can potentially cause irreversible damage to the developing brain [4]. Vitamin B12 is present only in foods of animal origin, and thus individuals consuming a diet with no or little animal-source or fortified foods are at increased risk of B12 deficiency.

Vitamin B12 status is assessed predominantly by direct measurement of total serum B12 concentration [5,6,7]. B12 is a cofactor for methionine synthase [8] and methylmalonyl CoA mutase [4], and because of this total homocysteine (tHcy) and methylmalonic acid (MMA), respectively, accumulate in individuals with suboptimal B12 status or deficiency [9]. Consequently, serum B12, alongside the functional biomarkers serum tHcy and/or MMA [5,6,7,10,11], is frequently used to evaluate vitamin B12 status. More recently a combined indicator of B12 status, summarizing the results of all three biomarkers, has been proposed (3cB12) [12,13]. Thresholds of 3cB12 to distinguish severe from marginal deficiency are not yet validated across populations and age groups hence public health decisions to supplement or treat currently rely on observed 3cB12 distribution and expert opinions [13].

The objective of the present study was to investigate B12 status before and after treatment with a standard or a reduced dose of RUTF and explore factors associated with the B12 status.

## 2. Materials and Methods

### 2.1. Study Design and Settings

This study was based on data from the Modeling an Alternative Nutrition Protocol Generalizable to Outpatient care (MANGO) trial for which the methodology is described in detail elsewhere [14]. In brief, MANGO was a randomized non-inferiority trial testing the efficacy of a reduced, compared with a standard, RUTF dose as recommended by WHO in the management of uncomplicated SAM in children aged 6–59 months, in Burkina Faso. The study was conducted from October 2016 to December 2018 in 10 health centers in the district of Fada N’Gourma, in the Eastern region, 220 km east of the capital, Ouagadougou. In 2016 the prevalence of wasting was 10% in the district [15] and in 2018 63% of children up to the age of 6 months were exclusively breastfed [16].

### 2.2. Study Participants

Children with SAM were admitted to the trial based on anthropometric entry criteria of a mid-upper arm circumference (MUAC) <115 mm, and/or weight-for-height z-score (WHZ) <−3 [17], without medical complications, and having passed a RUTF appetite test [18]. Before enrolment, all parents of eligible children with SAM received information concerning the study objectives and procedures, were informed of the possibility to withdraw at any time, and provided written consent to participate in the trial. Only children that had both admission and discharge serum samples available after analyses of iron and vitamin A status [19] were considered for the vitamin B12 sub-study.

### 2.3. Intervention

Treatment of children with SAM followed the Burkina Faso national Community-based Management of Acute Malnutrition (CMAM) guidelines [20] except for the RUTF dose. In brief, after obtaining caregiver consent, children were randomized individually to receive weekly either a standard dose of RUTF or a reduced dose of RUTF from the 3rd treatment week onwards until discharge. All other treatment was similar in both groups with systematic antibiotics given at admission, treatment of other childhood diseases diagnosed during study visits, and provision of measles vaccine if not given previously. Six, monthly vitamin A supplementation doses (100,000 IU for 6–11 month olds; and 200,000 IU for 12–59 month olds) were organized in the area during the study period. The RUTF used in the study were made of peanut butter, milk powder, vegetable oils, sugar, and a vitamin-mineral premix, packed in 92 g individual sachets providing approximately 500 kcal and 1.47 µg of B12 [21]. Caregivers in both groups were advised to offer family foods in addition to the prescribed RUTF if needed, and this was evaluated with a food intake survey at weeks 4 and 5 of treatment, after the shift in dosage in the reduced arm, as described previously [22,23]. The daily B12 dose according to the randomization arm and the child’s weight specifications [21] is described in Table 1.

Children were followed weekly until discharge and were classified as recovered, referred, non-response, defaulting, lost to follow-up, death, or false discharge [14]. Recovery was defined as reaching a WHZ of ≥−2 for those admitted with a WHZ < −3 only, or MUAC ≥ 125 mm for those admitted with a MUAC < 115 mm only, or both WHZ ≥ −2 and MUAC ≥ 125 mm for those admitted with both WHZ < −3 and MUAC < 115 mm, upon two consecutive visits and absence of any illness. Referrals concerned children who developed a medical complication, edema, or who experienced weight loss (>5% for 3 visits) or stagnation in weight gain (≤100 g over 4 weeks) during treatment; those children were referred to inpatient care. The maximum duration of treatment was 16 weeks. Non-response included children who had not recovered by 16 weeks of treatment. Defaulters were defined as those missing 3 consecutive visits, but the child could be confirmed alive. Lost to follow-up included children missing 3 consecutive visits, and the child could not be reached, or confirmed alive. False discharges included children who were erroneously discharged as recovered or referred, but upon analysis did not meet the discharge criteria [14]. A study nurse recorded complete medical history and performed morbidity assessment at admission and at each weekly visit thereafter. Fever was defined as an axillary temperature of ≥37.5 °C, which systematically resulted in a rapid malaria test, with a positive result defining malaria. Acute respiratory illness (ARI) was defined as a cough reported by the caregiver in the previous week or diagnosed by the study nurse during a consultation. Diarrhea included acute, persistent, or dysenteric forms and was defined as ≥3 loose stools per day as reported by the caregiver in the previous week or diagnosed by the study nurse. Iron deficiency anemia (IDA) was defined as hemoglobin (Hb) < 110 g/L and serum ferritin adjusted for inflammation (SF_adj_) of <12 μg/L as per Kangas et al. 2020 [20].

### 2.4. Data Collection

Socio-demographic characteristics were collected at admission, anthropometry and 2-week retrospective morbidity were collected at each visit [14]. Data were collected via tablets using Open Data Kit software V1.4.9 (https://www.kobotoolbox.org).

### 2.5. Blood Sampling and Storage

Venous blood was collected in Vacutainer^®^ (BD, Franklin Lakes, NJ, USA) clot activator tubes at admission and discharge from treatment. Blood samples were transported in a cold box at 2–8 °C to the field laboratory where samples were stored in a refrigerator at 2–10 °C for up to 24 h before processing. The blood samples were centrifuged at 3000 rotations per minute for 5 min (EBA 20 S Hettich, Germany) and serum was separated and stored at −20 °C. Available admission and discharge sera were stored at −20 °C for 3 to 5 years and then transported on dry ice in August 2021 from the collection site (Fada N’Gourma, Burkina Faso) to Bevital laboratory, Bergen, Norway (www.bevital.no). A total of 748 samples were analyzed for serum B12, tHcy, and MMA.

### 2.6. Biochemical Analyses

Serum B12 was determined with microbiological assay based on a colistin sulphate-resistant strain of *Lactobacillus leichmannii* [24]. The within- and between-day coefficient of variation was 5%. Serum tHcy and MMA were analyzed with gas chromatography-tandem mass spectrometry (GC-MS/MS) based on methylchloroformate derivatization [25]. For serum tHcy and MMA the within- and between-day coefficient of variation ranged from 1% to 3% and 1% to 5%, respectively. All B12 biomarkers were analyzed at Bevital laboratory. Serum ferritin (SF) and soluble transferrin receptor (sTfR) were previously determined using a combined sandwich-enzyme-linked immunosorbent assay [26] at VitMin Lab in Willstaedt, Germany when determining vitamin A and iron status [19]. Hemoglobin was measured in the field with a HemoCue^®^ 301 device (Hemocue AB, Ängelholm, Sweden).

### 2.7. Data Management

All preparation and statistical analyses were performed using Stata version 17 (Stata Corp, Texas, USA). Anthropometric z-scores were calculated using the 2006 World Health Organization (WHO) growth standards [27]. The combined indicator of vitamin B12 status (3cB12) was calculated to assess B12 status at admission and discharge and change in status according to the intervention arm. The 3cB12 was calculated by the log of serum B12 divided by the product of tHcy and MMA concentrations, (log10[(B12)/(MMAxtHcy)]) [12]. B12 status according to 3cB12 was classified as adequate if >0, moderate deficiency when ≥−1.4 but ≤0, marked deficiency when ≥−2.4 but <−1.4 and severe deficiency when <−2.4 [13].

### 2.8. Data Analyses

Population characteristics were reported as proportion, means ± standard deviation (SD), or median with Inter Quartile Range (IQR) (25th- and 75th-percentiles) as appropriate, with corresponding group size. Right-skewed outcomes were log-transformed prior to analysis and estimates were back-transformed. The baseline characteristics of children included in the B12 sub-study were compared to those who were not included.

Among children included in the B12 sub-study, the proportion of deficiencies at admission and discharge were calculated according to 3cB12 thresholds presented above. T-tests were used to investigate the effect of the intervention on mean biomarker concentrations by comparing the change from admission to discharge for all children combined, and presented as mean difference with 95% confidence intervals (CIs). Study site (health center) and research team were treated as a random effect, and we adjusted for age and sex at admission. False discharge children were not included in the analysis.

Linear mixed-effects modelling was used to investigate the effect of the intervention on mean biomarker concentrations, comparing the change from admission to discharge, according to RUFT group (standard vs. reduced dose) with adjustment for age, sex, and outcome measure at admission, and including study site and research team as random factors. Only samples from recovered children were used to estimate the effect of the RUTF dosage on B12 status.

Factors (age, gender, breastfeeding status, duration of treatment, acute illness, biochemical profile) potentially associated with B12 status (defined by 3cB12) in children with SAM at admission and discharge were explored using multivariate logistic mixed-effects regression [28]. The association between each independent variable and B12 status was assessed at admission and discharge in unadjusted and age-adjusted analyses. All statistical analyses were performing using Stata version 17 (Stata Corp, College Station, TX, USA) and *p*-values < 0.05 were considered significant.

### 2.9. Ethical Considerations

This study was approved by the National Ethics Committee of Burkina Faso (deliberation number 2015-12-00) and the national clinical trials board of Burkina Faso (Direction Générale de la Pharmacie, du Médicament et des Laboratoires (DGPML)). The trial was registered in the International Standard Randomized Controlled Trial Number (ISRCTN) registry as ISRCTN50039021. A second approval was obtained in October 2020 from the National Ethics Committee of Burkina Faso (N°2020/000148/MS/MESRSI/CERS) to analyze remaining sera for vitamin B12 analyses.

## 3. Results

### 3.1. Characteristics of Children in Vitamin B12 Sub-Study

Of the 801 children included in the MANGO trial, 374 (47%) were included in the B12 sub-study: 180 from the reduced and 194 from the standard RUTF arm (Figure 1). Thirteen children were excluded from the follow-up analyses due to being discharged as defaulted or false discharge, resulting in a total of 361 children (174 from reduced RUTF arm and 187 from standard RUTF arm) being included in the B12 status at discharge analyses.

There was no difference at baseline in demographic characteristics between the children included in the B12 biomarker sub-study compared to those not included from the MANGO cohort, and only minor differences in the occurrences of acute illnesses and serum ferritin concentrations (Table 2). Children included in the B12 sub-study had a median [IQR] age of 11.0 [7.7; 16.9] months, with 49% female. Approximately 85% (321/374) of children were breastfed at admission. Diarrhea or fever was observed in one quarter, and iron deficiency anemia in just over one-third of the children at admission.

### 3.2. Biomarkers of Vitamin B12 Status in Children with SAM Treated with RUTF

At admission, only 16% showed a 3cB12 value indicating adequate B12 status (3cB12 > 0), while 32% had 3cB12 values suggesting marked to severe B12 deficiency (3cB12 < −1.4) (Figure 2). After treatment the proportion of children with adequate B12 status increased from 16% to 36%; while the proportion with 3cB12 values suggesting marked to severe B12 deficiency decreased from 32% to 9% (Figure 2). All children at discharge who still had a marked or severe deficiency were still breastfed. 

The following cut-points for 3cB12 were employed for the classification of B12 status: adequate if >0, moderate deficiency when ≥−1.4 but ≤0, marked deficiency when ≥−2.4 but <−1.4, and severe deficiency when <−2.4 [13].

Treatment improved the B12 biomarker profile from intake to discharge by increasing serum B12 and decreasing tHcy, MMA, and 3cB12 (Table 3).

### 3.3. Vitamin B12 Biomarkers with a Reduced or a Standard RUTF Dose

No differences at admission or discharge in B12 biomarkers and response were observed according to intervention (a reduced or a standard RUTF dose) (Table 4).

### 3.4. Factors Associated with B12 Status

Factors or indicators potentially associated with vitamin B12 status, indicated by 3cB12, at admission and discharge are presented in Table 5. Breastfeeding was associated with B12 status at admission and discharge. Low vitamin B12 status was associated with children who were still breastfed both at admission and discharge. In contrast, non-breastfed children above 12 months of age (*n* = 51) showed a much higher 3cB12 value and were thereby judged to have a better B12 status than breasted children above 12 months (*n* = 169), both at admission and at discharge (*p-*value < 0.01). Amongst breastfed children, 34% were considered to have marked or severe B12 deficiency as judged by 3cB12 at admission, while for non-breastfed children, only 6% were classified as having marked or severe B12 deficiency. Following treatment of SAM, marked or severe B12 deficiency was no longer observed in non-breastfed children but persisted in 9% (*n* = 32) of breastfed children. Children who still had a marked or severe B12 status deficiency at discharge were all still breastfed.

At admission, children with diarrhea had higher B12 status (*p-*value < 0.01) than those without diarrhea. At discharge, children with iron deficiency anemia (IDA) showed a lower 3cB12 compared to children without IDA (*p-*value < 0.01). Length of treatment was not associated with 3cB12 (*p-*value > 0.05).

## 4. Discussion

To our knowledge, this study is the first to describe biomarkers of vitamin B12 status in children with SAM before and after outpatient treatment. Low concentrations of serum B12 alongside elevated serum tHcy and MMA were observed at admission with only 16% of children deemed to have adequate B12 status according to the 3cB12 index. Using 3cB12, 32% of children were judged to have marked to severe B12 deficiency at admission to treatment. Treatment with RUTF significantly reduced the proportion of children with marked to severe B12 deficiency, however, 9% of the children remained in this category at discharge. All non-breastfed children at discharge had adequate or moderate B12 status, and all children who still had a marked or severe B12 status deficiency at discharge (9%) were still breastfed (11% of breastfed children). Thus, while treatment of SAM, beyond correcting the anthropometric deficits, seems to be effective in improving vitamin B12 status, it appears to be insufficient to fully correct severe or marked deficiencies in some children.

The serum B12 levels we observed at admission in this study were lower than those found in healthy children aged 6–30 and 6–59 months in India [29,30] but similar to those found in children with moderate acute malnutrition (MAM) aged 6–23 in Burkina Faso treated with a 500 kcal/d lipid-based nutrient supplement (LNS) or corn soy blend (CSB) containing approximately 4.1 μg in CSB and 3.2 μg in LNS of vitamin B12 [31]. Notably, at discharge, children treated for MAM had on average lower serum B12 concentrations than children treated for SAM in the present study [31].

Functional consequences of a low B12 status are pernicious anemia and cognitive impairment as B12 is involved in myelination of neurons. These could explain the delay in cognitive and psychomotor development reported in children suffering from SAM [32]. Vitamin B12 has been linked to child neurodevelopment and associated with cognitive scores among school-aged children [33].

Up to 67% of children still presented with some degree of deficiency in vitamin B12 at discharge from SAM treatment. This raises the question of whether RUTF could be further fortified to improve vitamin B12 status at discharge. Because vitamin B12 is water-soluble and any excess is excreted in urine, the risk of reaching toxicity with higher fortification levels is unlikely [34]. Another way of improving vitamin B12 status of children with SAM could be via post-discharge supplementation with micronutrients. Considering the persistence of certain micronutrient deficiencies reported at the end of treatment [19], of which some might require longer time to normalize than the average malnutrition treatment preiod, the use of post-discharge supplementation strategies seems promising. Recently the use of SQ-LNS has been recommended to improve the nutritional status, health, and survival of children living in low-income settings where dietary quality is poor [35]. Additional fortification with vitamin B12 would add little cost to the fortification already in place for wheat and maize flour in sub-Saharan Africa.

Our study demonstrates that treatment with both standard and reduced RUTF improved the B12 biomarker profile in children with SAM. The Recommended Nutrient Intake (RNI) for vitamin B12 is 0.7–1.2 µg/d for children aged 6–59 months [36]. In the present study, the expected vitamin B12 intake from RUTF was 1.6–6.3 µg/d for children receiving the standard dose of RUTF and 1.5–2.9 µg/d for those offered the reduced dose. As weight gain was similar, most likely intake was similar. Thus, despite the reduction in the amount of RUTF offered during SAM treatment, the expected intake for vitamin B12 was still above the RNI for non-malnourished children. Further, the estimated maximum capacity for active absorption of vitamin B12 is approximately 1.5–2 µg per meal/dose for adults [37,38]. In other words, the expected B12 absorption was close to the available B12 in RUTF irrespective of whether a child was randomized to receive a standard or reduced RUTF dose. In addition, in this study, complementary and family foods contributed to at least 9% of daily B12 intakes during outpatient SAM treatment [22]. While the dose of RUTF offered to children depended on the randomization arm of the child, we have previously reported data on sources of nutrient intakes from complementary and family foods which indicated B12 intake was similar for the two groups [22].

Deficiencies in B12 were markedly higher in young and breastfed children, under 2 years. This was also found in previous studies [29,31]. As the youngest children are typically the children most likely to be breastfed, there is often confusion around the contribution of age and breastfeeding to B12 status, and the two factors are difficult to completely disentangle. In our study, B12 status was associated with breastfeeding status. This could be due to inadequate B12 status of the mother translating to low B12 concentration in breastmilk [39,40,41]. In fact, B12 concentrations in human milk are highest in colostrum and decrease during the lactation period [42]. It is estimated that human milk of women with adequate vitamin B12 status provides 0.4 µg of vitamin B12 per day [43,44]. Therefore, an extra 0.4 µg/day of vitamin B12 is needed during lactation in addition to the normal adult requirement of 2.0 µg/day, giving a total Estimated Average Requirement of 2.4 µg/day and a RNI of 2.8 µg/day for lactating women [36]. High rates of B12 deficiency in pregnant and breastfeeding women and their infants have repeatedly been demonstrated in developing countries [45]. In Guatemala, Anaya-Loyola et al. [46] found that breast milk provides inadequate amounts of B12 for the predominantly breasted infant. A similar difference in serum B12 between breastfed and non-breastfed children (183 versus 334 pmol/L) was reported from a large study among 6- to 30-month old North Indian children [47]. Maternal dietary data were not available, but consumption of animal source or fortified foods was considered low [47]. With potentially inadequate vitamin B12 status and intake among lactating women, breastmilk B12 content could be compromised, increasing the risk of low B12 status among breastfed infants. A lower B12 status in breast-fed children as compared to non-breast fed children has previously been described [45].

No diarrhea at admission and iron deficiency anemia at discharge were associated with lower B12 status. This is in line with previous studies among 6- to 35-month old Nepalese children [48] but not confirmed for children with diarrhea in the present study. In India, Kumar et al., (2018) also reported that vitamin B12 deficiency was low in children with persistent diarrhea, thrombocytopenia, and anemia [49]. Indeed, the sample of children with diarrhea is very small, only 14 children. This could explain the opposite results obtained. 

Treatment length appeared to have no impact on 3cB12. The lack of association between duration of treatment and change in B12 status would suggest low B12 status was corrected soon after treatment commenced and thereafter, the B12 biomarker concentrations stabilized over the course of the treatment period. However, with only admission and discharge blood sampling we can only speculate and are unable to investigate time to B12 repletion more thoroughly.

### 4.1. Strengths and Limitations

The original MANGO study was not designed to investigate changes in vitamin B12 status specifically. However, this sub-study included three individual biomarkers of B12 status as well as a composite score (3cB12). The 3cB12 composite score is considered to give a better judgment of B12 status than use of the individual biomarkers [26]. We also analyzed a rather large sample of children (*n* = 374) compared to many of the studies in this field, and the individual randomization of children was a strength of this study.

One of the limitations of this study is that samples were stored for between 3 and 5 years prior to analyses at a non-optimal, −20 °C, temperature. For serum vitamin B12, an almost linear decrease has been observed during the time of storage at −80 °C [50]. In a previous study, the mean level of serum vitamin B12 decreased from 100% to 85.6%, 62.2%, and 50.5% at time points 1, 8, and 13 years, respectively, when stored at −80 °C [50]. However, another study demonstrated that serum cobalamin, tHcy, and MMA are stable when stored at −25 °C for almost 30 years [51]. Because all our samples were stored for the same amount of time, we believe that the main findings of significant change in response to treatment and no difference in endline status according to RUTF dose are robust. Folate was not measured in this study, and thus we cannot judge the need for supplementation with this vitamin [13]. Lastly, our efforts to disentangle and more robustly investigate the association between age and breastfeeding in relation to B12 status were impaired by the small number of non-breastfed children in this study. These results should thus be interpreted with caution.

### 4.2. Generalizability

Our findings are, to a certain extent, generalizable to children with SAM of comparable age in other low-income countries with similar socioeconomic conditions, and where the food habits are comparable. Food groups most consumed by children in our study were grains, roots or tubers (96%), and legumes and nuts (72%) [52]. Egg consumption was low (3%) [52]. Most of the children in this study were young (less than 2 years old) and still breastfed. Therefore, we are uncertain whether our findings on the prevalence of low B12 status can be generalized to healthy children from a similar LMIC setting, or to children with SAM above 2 years of age who are not predominantly breastfed.

## 5. Conclusions

Children with SAM presented with a low B12 status, as judged by 3cB12, and generalized deficiencies (IDA and anemia) at admission to treatment. However, treatment with RUTF significantly improved B12 status though it did not normalize 3cB12 in all children. Reduction of the RUTF dose did not impact the final status of B12 biomarkers. Up to 67% of young and breast-fed children still showed B12 status deficiency as judged by 3cB12 at the end of the treatment. More research is needed on the potential benefits of higher levels of RUTF fortification or dietary improvement strategies to increase B12 intake during treatment, or of supplementation after treatment of SAM to normalize B12 status.

## Figures and Tables

**Figure 1 nutrients-15-03496-f001:**
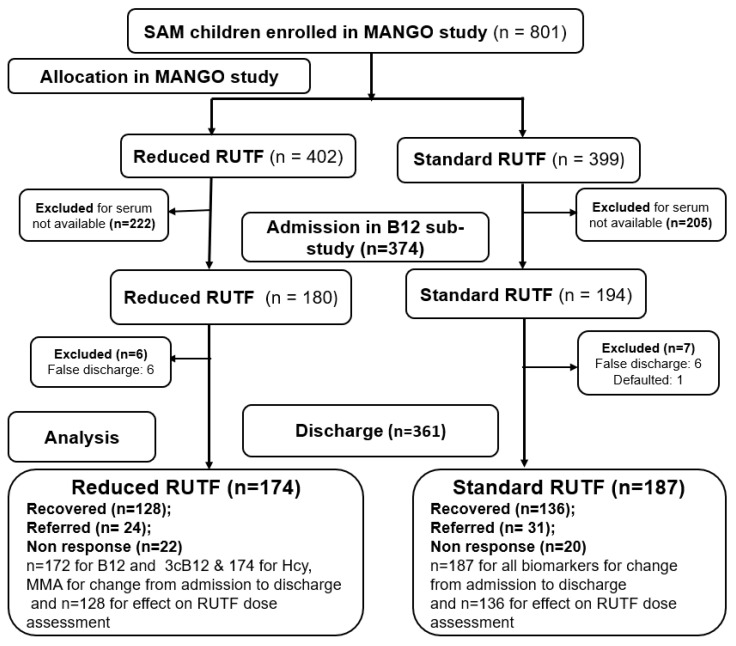
Inclusion of participants from the MANGO trial [12] to the vitamin B12 sub-study.

**Figure 2 nutrients-15-03496-f002:**
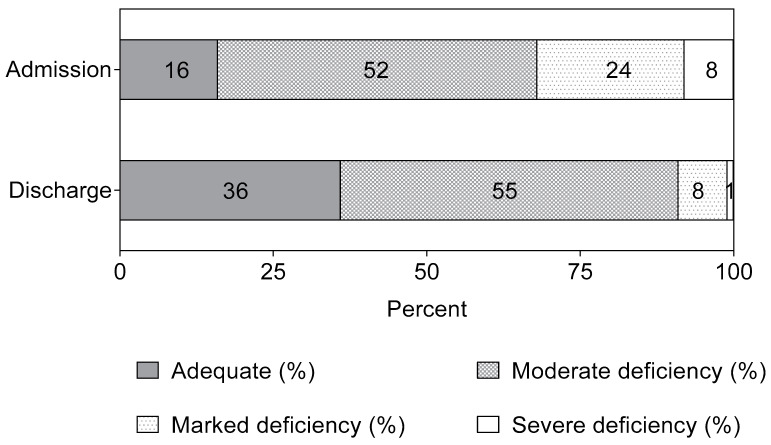
Distribution (%) of B12 status in children with SAM at admission (*n* = 374) and discharge following treatment (*n* = 359) as judged by 3cB12.

**Table 1 nutrients-15-03496-t001:** Daily standard and reduced doses of RUTF with respective vitamin B12 content.

Childs Weight (kg)	Standard RUTF Dose		Reduced ^1^ RUTF Dose	
	RUTF Quantity/Day (Sachets)	B12 Content of Daily RUTF Dose (µg)	RUTF Quantity/Day (Sachets)	B12 Content of Daily RUTF Dose (µg)
3.0–3.4	1.1	1.62	1.0	1.47
3.5–4.9	1.4	2.06	1.0	1.47
5.0–6.9	2.1	3.09	1.0	1.47
7.0–9.9	2.9	4.26	2.0	2.94
10.0–14.9	4.3	6.32	2.0	2.94

RUTF, ready-to-use therapeutic foods. ^1^ Reduced dose arm received a standard dose of RUTF for the first 2 weeks, then the reduced dose from the 3rd week onwards.

**Table 2 nutrients-15-03496-t002:** Characteristics of children with SAM aged 6–59 months at admission from the MANGO study [12] included in the present B12 sub-study, compared with those excluded.

Variables	*n*	Excluded (*n* = 427)	Included (*n* = 374)	*p-*Value ^2^
Age (months)	801	9.5 [7.2; 15.9]	11.0 [7.7; 16.9]	0.34
Female	801	52 (221)	49 (184)	0.47
Breastfeeding	801	86 (368)	86 (321)	0.89
Weight (kg)	801	6.2 (1.3)	6.3 (1.3)	0.09
Height (cm)	801	68.8 (7.7)	69.5 (7.7)	0.25
MUAC (mm)	801	113 (6.8)	113 (6.1)	0.14
Acute illnesses % (*n*)				
Malaria	801	29 (125)	37 (138)	0.02
Acute respiratory illness	801	36 (153)	26 (98)	<0.01
Diarrhea	801	25 (105)	25 (95)	0.79
Fever	801	27 (116)	25 (92)	0.41
Biochemical profile				
Hemoglobin (g/L)	801	97.0 [84.0; 108]	99.0 [86.0; 108]	0.14
<110 g/L	801	79 (335)	78 (293)	0.97
Serum ferritin (µg/L)	754 ^1^	14.2 [6.6; 32.3]	15.9 [7.8; 30.6]	0.01
<12 µg/L	754 ^1^	43 (164)	40 (150)	0.58
Serum soluble transferrin receptor (mg/L)	754 ^1^	8.3 [6.1; 12.8]	8.5 [6.6; 12.8]	0.24
>8.3 mg/L	754 ^1^	50 (190)	52 (194)	0.31
Iron deficiency anemia	754 ^1^	39 (147)	39 (144)	0.83

^1^ Analyzed for 380 children excluded from the current sub-study and 374 children included in the B12 sub-study. Data shown are % (*n*), mean (± SD) or median [IQR]. ^2^ Difference analyzed by using *t*-test.

**Table 3 nutrients-15-03496-t003:** Vitamin B12 biomarkers at admission to and discharge from treatment for children with SAM aged 6–59 months.

Serum Biomarkers	Admission (*n* = 374)		Discharge (*n* = 361)		Difference ^2^	
	*n*	Median [IQR]	*n*	Median [IQR]	Mean [95% CI]	*p* Value
B12, pmol/L	374	188 [125; 266]	359 ^1^	279 [207; 365]	66 [55; 77]	<0.0001
tHcy, µmol/L	374	11 [7; 15]	361	8 [6; 10]	−4 [−4.4; −3.2]	<0.0001
MMA, µmol/L	374	0.59 [0.31; 1.12]	361	0.37 [0.25; 0.73]	−0.26 [−0.36; −0.17]	<0.0001
3cB12	374	−1 [−1.6; −0.4]	359 ^1^	−0.3 [−0.9; 0.2]	0.5 [0.4; 0.6]	<0.0001

3cB12: combined indicator of vitamin B12 status; B12: serum cobalamin; IQR, interquartile range MMA: methylmalonic acid; tHcy: total homocysteine. ^1^ Two children were excluded from B12 biomarker analyses (and thereby 3cB12) due to low sample volumes. ^2^ Difference analyzed by using T-test with study site and research team as random effects and adjusting for sex, age.

**Table 4 nutrients-15-03496-t004:** Vitamin B12 biomarkers at admission and discharge among children recovered from SAM and treated with a reduced or a standard RUTF.

Outcomes	Admission				Recovery			
	Reduced RUTF (*n* = 180)	Standard RUTF (*n* = 194)	Difference [95% CI] ^1^	*p-*Value	Reduced RUTF (*n* = 128)	Standard RUTF (*n* = 136)	Difference [95% CI] ^2^	*p-*Value
B12 (pmol/L)	198 [124; 286]	174 [125; 251]	−17 [−40; 5]	0.14	290 [209; 363]	267 [193; 350]	−6 [−29; 17]	0.61
tHcy (µg/L)	11 [8; 15]	11 [7; 16]	0.6 [−0.8; 2.0]	0.39	8 [6; 10]	8 [6; 10]	0.1 [−0.5; 0.8]	0.68
MMA (µmol/L)	0.71[0.33; 1.22]	0.56 [0.31; 1.00]	−0.10 [−0.30; 0.11]	0.35	0.41 [0.24; 0.71] ^3^	0.37 [0.25; 0.76]	0.05 [−0.07; 0.16]	0.42
3cB12	−1.1 [−1.7; −0.4]	−0.9[−1.5; −0.4]	0.01 [−0.16; 0.18]	0.93	−0.27 [−0.82; 0.24] ^3^	−0.40 [−0.90; 0.14]	−0.07 [−0.20; 0.05]	0.26

3cB12: combined indicator of vitamin B12 status; B12: serum cobalamin; IQR, interquartile range MMA: methylmalonic acid; tHcy: total homocysteine. Data are median [IQR]. ^1^ Difference analyzed by using linear mixed-effects models with adjustment for age and sex and including study site and research team as random factors. ^2^ Difference analyzed by using linear mixed-effects models with adjustment for age and sex and outcome measure at admission and including study site and research team as random factors. ^3^
*n* = 127 for reduced RUTF.

**Table 5 nutrients-15-03496-t005:** Associated factors of 3cB12 status at admission and discharge to treatment in unadjusted and age-adjusted models for children aged 6–59 months with SAM.

Characteristics	Admission (*n* = 374)					Discharge (*n* = 361)				
	3cB12, Median [IQR]	Unadjusted		Adjusted ^1^		3cB12, Median [IQR]	Unadjusted		Adjusted ^1^	
Child’s Sex		β	*p*-Value	β	*p*-Value		β	*p*-Value	β	*p*-Value
Male	−0.91 [−1.63; −0.43]	Ref		Ref		−0.35 [−0.85; 0.08]	Ref		Ref	
Female	−1.05 [−1.63; −0.23]	0.01	0.94	0.05	0.53	−0.34 [−0.87; 0.28]	0.02	0.79	0.06	0.44
Breastfeeding status										
No	−0.18 [−0.75; 0.47]	Ref		Ref		0.19 [−0.31; 0.58]	Ref		Ref	
Yes	−1.10 [−1.70; −0.48]	−0.91	<0.001	−0.51	<0.01	−0.44 [−0.95; 0.07]	−0.57	<0.001	−0.33	0.01
Malaria										
No	−0.92 [−1.64; −0.44]	Ref		Ref		−0.36 [−0.86; 0.18]	Ref		Ref	
Yes	−1.06 [−1.56; −0.18]	0.14	0.16	0.00	0.99	−0.18 [−0.84; 0.29]	0.05	0.78	0.08	0.62
Acute Respiratory illness										
No	−0.99 [−1.66; −0.40]	Ref		Ref		−0.32 [−0.85; 0.19]	Ref		Ref	
Yes	−0.92 [−1.45; −0.34]	0.18	0.10	0.15	0.14	−0.49 [−1.06; 0.10]	−0.07	0.70	−0.05	0.77
Diarrhea										
No	−1.10 [−1.68; −0.45]	Ref		Ref		−0.36 [−0.86; 0.18]	Ref		Ref	
Yes	−0.69 [−1.33; −0.11]	0.29	0.01	0.24	0.01	−0.11 [−0.34; 0.26]	0.23	0.25	0.27	0.16
Fever										
No	−0.98 [−1.68; −0.37]	Ref		Ref		−0.36 [−0.87; 0.19]	Ref		Ref	
Yes	−0.94 [−1.37; −0.42]	0.12	0.27	0.09	0.37	−0.27 [−0.47; 0.20]	0.09	0.61	0.05	0.78
Hemoglobin										
<110 g/L	−0.80 [−1.46; −0.28]	Ref		Ref		−0.24 [−0.72; 0.20]	Ref		Ref	
≥110 g/L	−1.01 [−1.63; −0.41]	−0.09	0.45	−0.08	0.47	−0.49 [−0.95; 0.16]	−0.10	0.21	−0.11	0.16
Ferritin										
<12 µg/L	−0.96 [−1.58; −0.40]	Ref		Ref		−0.31 [−0.74; 0.21]	Ref		Ref	
≥12 µg/L	−0.97 [−1.68; −0.38]	−0.03	0.79	−0.02	0.86	−0.38 [−1.00; 0.12]	−0.10	0.20	−0.08	0.25
Soluble transferrin receptor (STfR)										
<8.3 mL/L	−1.07 [−1.61; −0.44]	Ref		Ref		−0.32 [−0.82; 0.19]	Ref		Ref	
≥8.3 mL/L	−0.89 [−1.63; −0.29]	0.05	0.57	0.06	0.50	−0.36 [−0.87; 0.16]	−0.04	0.63	−0.02	0.77
Iron deficiency anemia (IDA)										
No	−0.93 [−1.55; −0.40]	Ref		Ref		−0.30 [−0.74; 0.19]	Ref		Ref	
Yes	−0.97 [−1.72; −0.38]	−0.06	0.55	−0.05	0.57	−0.69 [−1.18; 0.15]	−0.21	0.02	−0.20	0.03
Length of treatment										
<8 weeks	−0.93 [−1.63; −0.34]	Ref		Ref		−0.30 [−0.85; 0.16]	Ref		Ref	
≥8 weeks	−1.01 [−1.61; −0.41]	−0.03	0.74	0.10	0.25	−0.37 [−0.86; 0.19]	0.02	0.84	0.06	0.40

β; standardized coefficient (ratio of standard deviation), unitless, that allows independent comparison from variables tested in the model, to quantify the “magnitude” of the effect of one variable on another. The higher β, the stronger the effect of the variable tested. ^1^ Mixed regression models adjusting for age and including study site and research team as random factors.

## Data Availability

Data of this study has been attached as a Appendix A file.

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
