# Peer review of "Vitamin B12 Status before and after Outpatient Treatment of Severe Acute Malnutrition in Children Aged 6–59 Months: A Sub-Study of a Randomized Controlled Trial in Burkina Faso"

_nutrients, 2023, doi:10.3390/nu15163496_

Round 1

Reviewer 1 Report

The article of Nikiema and colleagues evaluated B12 status in children under five in Burkina Faso with severe acute malnutrition before and after treatment with a standard or a reduced dose of RUTF and explore factors associated with the B12 status. The reviewer commends the authors for the intricate amount of work performed, especially for B12 analyses in several samples. The manuscript read easily and it was a pleasure to learn more about the use of the different metrics for B12 deficiency, especially with children. The following are comments to strengthen this manuscript.

Ln 126. What is the purpose of this sentence? Was anemia measured? If so, how?

Ln 131. Remove typo s

Ln 261. The diarrhea results and numbers in Table 5 don’t match. At admission, children with diarrhea had higher B12 status than those without diarrhea.

Ln 262. The description of the results for anemia is a bit confusing. Are you talking about at admission or at discharge?

Ln 293 and on. the comments on adding more B12 to RUTFs or to common foods is well taken. You might want to add that this additional vitamin would add little cost to the fortification mechanisms for wheat and maize flour currently in place in SSA.  

Ln 328. The units might be incorrect.

Ln 341. Check the relationship between diarrhea and b12 status at admission. Based on Table 5, kids with diarrhea at admission had higher b12 status than those without diarrhea.

Ln 321-340. This paragraph explains why B12 mothers might not be able to address the B12 needs of their infants through breast milk, especially during exclusive breastfeeding. However, how do you explain the higher, though not adequate, B12 levels seen in non-breastfed children? Based on your further comments, this might not be possible as consumption of animal sources is limited among adults and children.

Limitations section. You might want to add that folate was not measured in this study. This is because the authors of reference [13] discussed the inclusion of folate for correction purposes, especially when folate is low.

Ln 421. Please correct the statement.

Author Response

Reviewer 1

Comments and Suggestions for Authors

The article of Nikiema and colleagues evaluated B12 status in children under five in Burkina Faso with severe acute malnutrition before and after treatment with a standard or a reduced dose of RUTF and explore factors associated with the B12 status. The reviewer commends the authors for the intricate amount of work performed, especially for B12 analyses in several samples. The manuscript read easily, and it was a pleasure to learn more about the use of the different metrics for B12 deficiency, especially with children. The following are comments to strengthen this manuscript.

 Authors: Thanks.

Ln 126. What is the purpose of this sentence? Was anemia measured? If so, how?

Authors: We replaced line 12- by: “Iron deficiency anemia (IDA) was defined as hemoglobin (Hb) < 110 g/L and serum ferritin adjusted for inflammation (SFadj) of <12 μg/L. as per Kangas et al. 2020 [20]”. (Lines 126-127).

Ln 131. Remove typo s

Authors: It’s done.

Ln 261. The diarrhea results and numbers in Table 5 don’t match. At admission, children with diarrhea had higher B12 status than those without diarrhea.

Authors: Exact. We have revised the data and corrected the text: “At admission, children with diarrhea had higher B12 status (P-value < 0.01) than those without diarrhea”. (Lines 255-256)

Ln 262. The description of the results for anemia is a bit confusing. Are you talking about at admission or at discharge?

Authors: We're talking about the discharge. So we've corrected the sentence: "At discharge, children with iron deficiency anaemia (IDA) showed a lower 3cB12 compared to children without IDA (P-value < 0.01)”. (Lines 256-257)

Ln 293 and on. the comments on adding more B12 to RUTFs or to common foods is well taken. You might want to add that this additional vitamin would add little cost to the fortification mechanisms for wheat and maize flour currently in place in SSA.  

Authors: It’s done. We added this sentence: “Additional fortification with vitamin B12 would add little cost to the fortification already in place for wheat and maize flour in sub-Saharan Africa.” (Lines 297-299)

Ln 328. The units might be incorrect.

Authors: The units in the sentence (Line 331) refer to B12 concentration. It is expressed in pmol/L.

Ln 341. Check the relationship between diarrhea and b12 status at admission. Based on Table 5, kids with diarrhea at admission had higher b12 status than those without diarrhea.

Authors: We checked and corrected it. In fact, at admission, children with diarrhea had higher B12 status than those without diarrhea”. So we've replaced the part with the sentences below:

“No diarrhea at admission and iron deficiency anaemia at discharge were associated with lower B12 status. This is in line with previous studies among 6 to 35 months old Nepalese children [47] but not confirmed for children with diarrhea in the present study. In India, Kumar et al. (2018) also reported that vitamin B12 deficiency was low in children with persistent diarrhoea, thrombocytopenia and anaemia [48]. Indeed, the sample of children with diarrhoea is very small, only 14. This could explain the opposite results obtained.’’ (Lines 338-344)

Ln 321-340. This paragraph explains why B12 mothers might not be able to address the B12 needs of their infants through breast milk, especially during exclusive breastfeeding. However, how do you explain the higher, though not adequate, B12 levels seen in non-breastfed children? Based on your further comments, this might not be possible as consumption of animal sources is limited among adults and children.

Authors: We added this sentence: “A lower B12 status in breast-fed children as compared to non-breast fed children has previously been described [45].” (Lines 336-337).

Limitations section. You might want to add that folate was not measured in this study. This is because the authors of reference [13] discussed the inclusion of folate for correction purposes, especially when folate is low.

Authors: It’s done. We added: “Folate was not measured in this study, and thus we cannot judge the need for supplementation with this vitamin.” (Lines 366-368)

Ln 421. Please correct the statement.

Authors: We corrected with the right reference: “Stabler, S.P. Vitamin B 12 Deficiency. N. Engl. J. Med. 2013, 368, 149–160, doi:10.1056/NEJMcp1113996” (Line 429).

Reviewer 2 Report

This is a highly interesting paper, showing e.g., that breastfeeding does not prevent from B12 deficiency when the mothers are also  B12-deficient!

Lines 289-291: Known functional consequences of B12 deficiency are listed: What´s about such consequences in the present study population? If possible such defects should be listed in addition.

Author Response

REVIEWER 2:

Comments and Suggestions for Authors

This is a highly interesting paper, showing e.g., that breastfeeding does not prevent from B12 deficiency when the mothers are also B12-deficient!

Authors:  This may be the case, but we have not measured B12 status of the mothers in this study.

Lines 289-291: Known functional consequences of B12 deficiency are listed: What´s about such consequences in the present study population? If possible such defects should be listed in addition.

Authors: We added the sentence below: “Vitamin B12 has been show on child development and affects the cognitive scores of school-aged children”. (L285-286)
